# Targeting necroptosis in muscle fibers ameliorates inflammatory myopathies

Mari Kamiya[1], Fumitaka Mizoguchi[1,6], Kimito Kawahata[1,2], Dengli Wang[3], Masahiro Nishibori[3], Jessica Day [4], Cynthia Louis [4], Ian P. Wicks [4], Hitoshi Kohsaka[1,5] & Shinsuke Yasuda [1✉]

Muscle cell death in polymyositis is induced by CD8[+] cytotoxic T lymphocytes. We hypothesized that the injured muscle fibers release pro-inflammatory molecules, which would further accelerate CD8[+] cytotoxic T lymphocytes-induced muscle injury, and inhibition of the cell death of muscle fibers could be a novel therapeutic strategy to suppress both muscle injury and inflammation in polymyositis. Here, we show that the pattern of cell death of muscle fibers in polymyositis is FAS ligand-dependent necroptosis, while that of satellite cells and myoblasts is perforin 1/granzyme B-dependent apoptosis, using human muscle biopsy specimens of polymyositis patients and models of polymyositis in vitro and in vivo. Inhibition of necroptosis suppresses not only CD8[+] cytotoxic T lymphocytes-induced cell death of myotubes but also the release of inflammatory molecules including HMGB1. Treatment with a necroptosis inhibitor or anti-HMGB1 antibodies ameliorates myositis-induced muscle weakness as well as muscle cell death and inflammation in the muscles. Thus, targeting necroptosis in muscle cells is a promising strategy for treating polymyositis providing an alternative to current therapies directed at leukocytes.

[1] Department of Rheumatology, Graduate School of Medical and Dental Sciences, Tokyo Medical and Dental University (TMDU), Tokyo 113-8519, Japan. [2] Division of Rheumatology and Allergology, Department of Internal Medicine, St. Marianna University School of Medicine, Kawasaki, Kanagawa 216-8511, Japan. [3] Department of Pharmacology, Graduate School of Medicine, Dentistry, and Pharmaceutical Sciences, Okayama University, Okayama 700-8558, Japan. [4] The Walter and Eliza Hall Institute of Medical Research, Parkville, VIC 3052, Australia. [5] Present address: Chiba-Nishi General Hospital, Matsudo, Chiba 270-2251, Japan. [6] Deceased: Fumitaka Mizoguchi. ✉email: syasuda.rheu@tmd.ac.jp

Polymyositis (PM) is a chronic inflammatory myopathy, in which CD8+ cytotoxic T lymphocytes (CTLs) play a crucial role to induce muscle cell death. Proximal muscle weakness is the most common symptom leading to progressive and persistent disability. Current treatments of PM depend on non-specific immunosuppressants including glucocorticoids and immunosuppressive agents. However, some patients fail to respond to the immunosuppressive therapies, and some also suffer from infectious diseases during the treatment. In addition, muscle weakness persists in more than half of the patients despite reduction in muscle inflammation[1]. Therefore, a better therapeutic strategy that not only suppresses muscle inflammation but also prevents muscle weakness and at the same time avoiding an increasing risk of infection is needed.

The histopathological features in PM are necrotic and regenerating muscle fibers as well as mononuclear inflammatory cell infiltrates including CD8+ T cells[2]. Analysis of peripheral blood of PM patients revealed clonally expanded CD8+ T cells, some of which were detected in the affected muscles[3]. CD8+ T cells in the muscles express cytotoxic effector molecules including perforin 1 (PRF1) and granzyme B (GZMB)[4]. PRF1 expressed by CD8+ T cells was distributed towards the adjacent muscle fibers, suggesting the involvement of PRF1 in the injury of muscle fibers[5]. Indeed, we previously showed that deleting PRF1 reduced the severity and incidence of myositis in C protein–induced myositis (CIM), a mouse model of PM[6]. However, the suppressive effects of PRF1 deficiency were partial, implying there are other pathways besides the PRF1 dependent one.

FAS ligand (FASLG) is another molecule that CD8+ CTLs can use to kill target cells. Histological examinations of PM muscle samples have shown that FAS and FASLG are expressed on both infiltrating lymphocytes and muscle fibers[7,8] without any signs of apoptosis in the muscle fibers[7]. Muscle fibers express CASP8 and FADD Like Apoptosis Regulator (CFLAR)[8], which suppresses the apoptosis pathway by inhibiting the activation of Caspase 8 (CASP8). These results suggest that the pattern of cell death in muscle fibers induced by CTLs is not apoptosis, and has been assumed to be necrosis instead[9].

Over the past decades, several types of cell death have been identified. Necroptosis is a genetically regulated form of lytic cell death, in which the morphological features are shared with necrosis[10]. During the impairment of apoptosis pathway, stimulation of death receptors including FAS activates Receptor Interacting Serine/Threonine Kinase 1 (RIPK1), followed by the activation of RIPK3 and Mixed Lineage Kinase Domain Like Pseudokinase (MLKL), which induces necroptosis of the cells[11]. The necroptotic cells subsequently release inflammatory molecules including damage-associated molecular patterns (DAMPs) and cytokines, which cause tissue inflammation.

In this study, we show that the pattern of cell death of muscle fibers in PM is FASLG-dependent necroptosis, while that of satellite cells and myoblasts is PRF1/GZMB-dependent apoptosis, using human muscle biopsy specimens of PM patients and models of PM in vitro and in vivo. Inhibition of necroptosis ameliorated myositis-induced muscle weakness as well as muscle cell death and inflammation in the muscles.

## Results

**Cell death of muscle fibers is necroptotic in PM.** To determine the pattern of cell death in PM, we first evaluated the presence of terminal deoxynucleotidyl transferase nick-end labeling (TUNEL) positive cells in the muscle specimens of PM patients. Histological analysis revealed that TUNEL positive cells were observed around muscle fibers (Fig. 1a, Supplementary Fig. 1a, b). Most of the TUNEL positive cells expressed Paired Box 7 (PAX7), which is a marker of satellite cells. In contrast, TUNEL positive muscle fibers were not observed in all of the specimens ($n = 9$ donors). These results indicate that the pattern of cell death in satellite cells is apoptosis, while that in muscle fibers is not. Next, we examined whether the dying muscle fibers express the molecular markers for necroptosis. Immunofluorescence staining revealed that the dying muscle fibers, which are identified as cells with reduced eosin staining in the cytoplasm, expressed RIPK1, RIPK3, MLKL and phosphorylated MLKL, whereas the expression levels of these proteins in the intact muscle fibers were low or absent (Fig. 1b, Supplementary Fig. 1c–e, g). We found some of the dying muscle fibers expressed phosphorylated MLKL, which localizes on the plasma membrane upon necroptosis[12–14]. Some of such muscle fibers expressed phosphorylated MLKL throughout cytoplasm while others did markedly on their plasma membrane, suggesting there could be variations in its localization among the muscle fibers depending on the execution status of necroptosis. While expression of CASP8 was observed in both PAX7 positive satellite cells and dying muscle fibers, expression of the active 18 kDa CASP8 subunit was only observed in the satellite cells (Fig. 1c, Supplementary Fig. 1h, i). Furthermore, the dying muscle fibers expressed high levels of FAS and CFLAR, an antiapoptotic protein capable of suppressing CASP8 activation (Fig. 1b, Supplementary Fig. 1c–e). These results suggest that the dying muscle fibers in PM are necroptotic. We also examined the muscle specimens of dermatomyositis (DM), which is another subset of idiopathic inflammatory myopathy, for the pattern of cell death of muscle fibers. Consistent with the findings in PM, most of the TUNEL positive cells were satellite cells and TUNEL positive muscle fibers were not observed (Supplementary Fig. 1b) in any of the specimens ($n = 3$ donors). The dying muscle fibers expressed high levels of molecular markers for necroptosis including FAS and CFLAR (Supplementary Fig. 1e, f) while the expression of active 18 kDa CASP8 subunit was observed only in satellite cells and inflammatory infiltrates (Supplementary Fig. 1i). These results suggest that the dying muscle fibers in DM also undergo necroptosis.

**FAS-FASLG pathway is crucial for CTLs to induce myotube death.** To investigate the involvement of apoptosis and necroptosis pathways in the antigen-dependent CTL-induced muscle cell death, we employed an in vitro model of PM where OT-I CTLs were co-cultured with C2C12 cell-derived myoblasts and myotubes that were retrovirally transduced with the genes encoding MHC class I (H2Kb) and SIINFEKL peptide derived from ovalbumin (OVA)[15]. In this model, H2KbOVA-transduced myoblasts (H2KbOVA-myoblasts) and myotubes (H2KbOVA-myotubes) are considered as satellite cells and muscle fibers in vivo, respectively[16,17]. We first tested whether PRF1 and GZMB are involved in CTL-induced muscle cell death. We found that deleting PRF1 or GZMB in OT-I CTLs reduced the cytotoxicity of OT-I CTLs against H2KbOVA-myoblasts (Fig. 2a). In contrast, absence of PRF1 or GZMB did not suppress CTL-induced cell death of H2KbOVA-myotubes (Fig. 2b, c). Next, we examined the role of FAS and FASLG in CTL-induced muscle cell death. Blockade of FASLG with a FAS-Fc chimeric protein did not reduce the cytotoxicity of OT-I CTLs against H2KbOVA-myoblasts (Fig. 2d), but reduced the cytotoxicity against H2KbOVA-myotubes (Fig. 2e). These results indicate that the PRF1/GZMB pathway is involved in CTL-induced cell death in myoblasts while the FAS-FASLG pathway plays a crucial role in CTL-induced cell death in myotubes.

**Cell death of myotubes was non-apoptotic.** Since FAS-FASLG pathway can drive both apoptosis and necroptosis, we

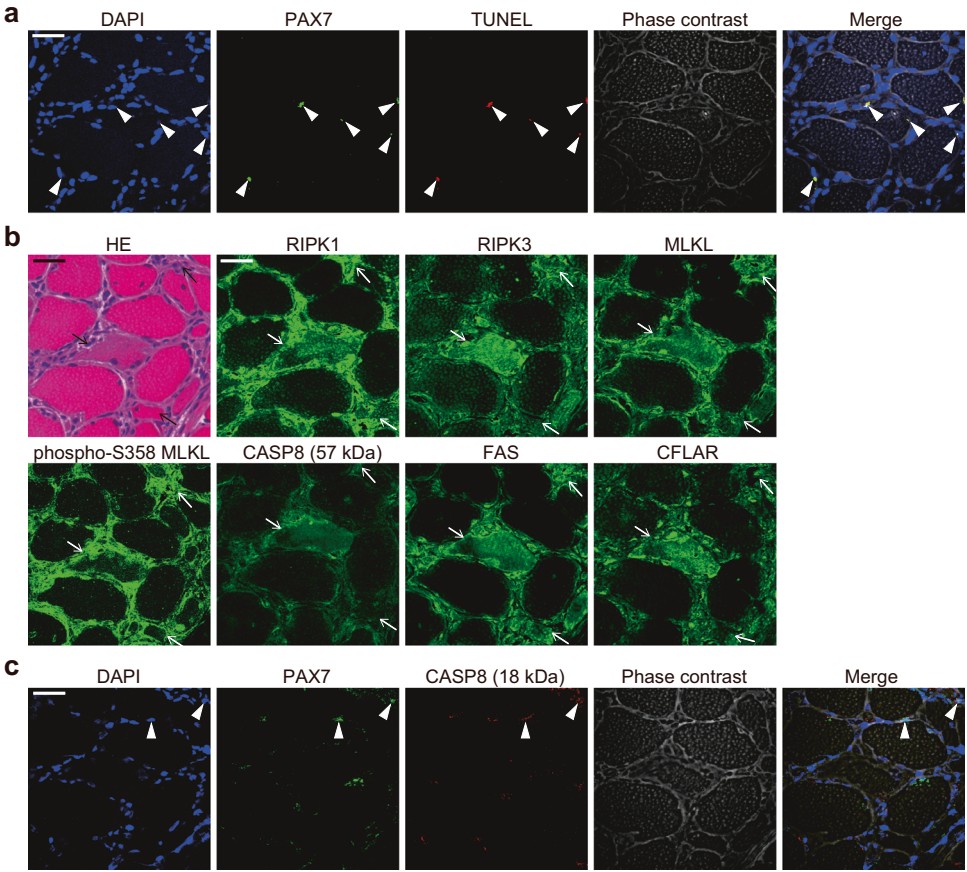

**Fig. 1 Expression of necroptosis-associated proteins in dying muscle fibers in PM. a–c** Representative images of muscle specimens of PM patients ($n = 9$). Scale bar indicates 20 μm. **a** Immunofluorescence staining against PAX7 (green) and the TUNEL staining (red). Nuclei were counterstained with DAPI (blue). Arrowheads indicate TUNEL positive PAX7 positive satellite cells. **b** Hematoxylin & Eosin (HE) and immunofluorescence staining against RIPK1, RIPK3, MLKL, phosphorylated MLKL at S358 (phospho-S358 MLKL), CASP8 (57 kDa), FAS, and CFLAR (green). The arrows indicate the dying muscle fibers, which showed reduced eosin staining in the cytoplasm. **c** Immunofluorescence staining against PAX7 (green) and active 18 kDa CASP8 subunit (red). Nuclei were counterstained with DAPI (blue). Arrowheads indicate active 18 kDa CASP8 subunit positive PAX7 positive satellite cells.

examined the type of cell death of myotubes by CTLs using time-lapse imaging of cell death visualized by Annexin V and propidium iodide (PI). H2K^bOVA-myoblasts co-cultured with OT-I CTLs stained positive for Annexin V before they became PI positive, indicating that they are apoptotic (Fig. 3a). Annexin V positive and PI negative H2K^bOVA-myotubes were not observed in the co-culture, but Annexing V negative and PI positive myotubes were observed instead (Fig. 3b). In TUNEL staining, the TUNEL positive cells were observed in 46 out of 54 myoblasts (85%) co-cultured with OT-I CTLs (Fig. 3c). On the contrary, none of the myotubes were TUNEL positive in the co-culture with OT-I CTLs (Fig. 3d). These results indicate that the CTL-induced cell death of myoblasts and myotubes is apoptosis and non-apoptosis, respectively. Immunofluorescence staining revealed that the co-cultured myotubes expressed FAS, CFLAR, and necroptosis-associated molecules including RIPK1, RIPK3, MLKL, and phosphorylated MLKL[18], suggesting involvement of FAS-mediated necroptosis in CTL-induced death of myotubes (Fig. 3e).

**Cell death of myoblasts was apoptosis and that of myotubes was necroptosis.** To confirm that CTL-induced myotube death is not apoptosis but necroptosis, we examined the effect of z-VAD-fmk, a pan-caspase inhibitor, and Necrostatin-1s (Nec1s), a necroptosis inhibitor targeting RIPK1 kinase, on CTL-induced

muscle cell death in vitro. While CTL-induced myotube death was not suppressed by z-VAD-fmk (Fig. 4a), it was suppressed by Nec1s in a dose-dependent manner (Fig. 4b). In contrast, CTL-induced myoblast death was suppressed by z-VAD-fmk (Fig. 4c), but not by Nec1s (Fig. 4d). The suppressive effect on CTL-induced myotube death was also observed by silencing of *Ripk3* with small interfering RNA (siRNA) in the myotubes (Fig. 4e). Silencing of *Ripk3* did not affect the fusion of myoblasts (Supplementary Fig. 2a), the number of nuclei in the myotubes (Supplementary Fig. 2b), nor the expression of myogenin (MYOG) and FAS (Fig. 4f), indicating that the suppressive effect on CTL-induced myotube death by silencing of *Ripk3* was not due to the inhibition of the differentiation to myotubes or decreased expression of FAS on the myotubes. These results confirmed that the cell death of myoblasts was apoptosis and that of myotubes was necroptosis.

**Inhibition of necroptosis suppressed CTL-induced release of inflammatory mediators from myotubes.** Since necroptosis is an inflammatory form of cell death, we next studied the effect of necroptosis inhibition on CTL-induced release of inflammatory mediators from myotubes. While the levels of High Mobility Group Box 1 (HMGB1), which is one of the family of DAMPs, and inflammatory cytokines including IL-1α and IL-6, were not detectable in the supernatants of mono-cultured

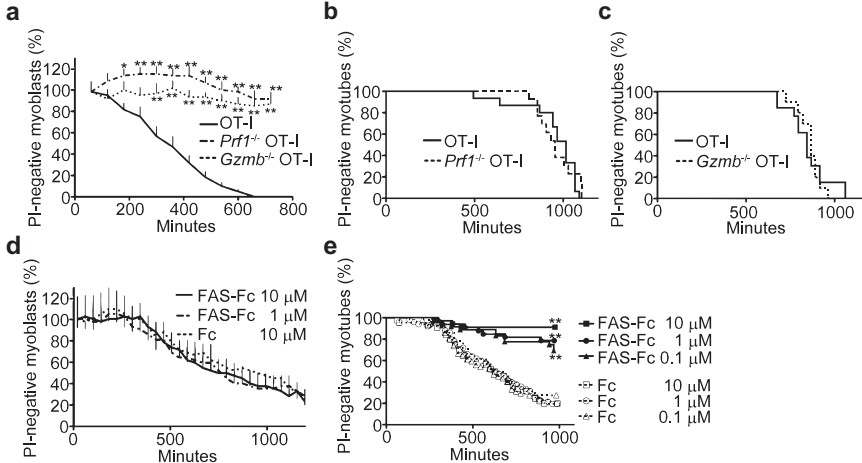

**Fig. 2 Involvement of PRF1, GZMB, and FAS/FASLG in CTL-induced cell death of myoblasts and myotubes. a** The viability of H2K$^b$OVA-myoblasts co-cultured with OT-I CTLs ($n = 80$), $Prf1^{-/-}$ OT-I CTLs ($n = 59$), or $Gzmb^{-/-}$ OT-I CTLs ($n = 61$). The viability was visualized by PI staining and assessed with time-lapse imaging. Data are presented as mean and SD. Two-way analysis of variance (ANOVA) test, followed by Dunnett's test. *$p < 0.05$, **$p < 0.01$. **b** The viability of H2K$^b$OVA-myotubes co-cultured with OT-I CTLs ($n = 30$) or $Prf1^{-/-}$ OT-I CTLs ($n = 26$). **c** The viability of H2K$^b$OVA-myotubes co-cultured with OT-I CTLs ($n = 42$) or $Gzmb^{-/-}$ OT-I CTLs ($n = 40$). **d** The viability of H2K$^b$OVA-myoblasts co-cultured with OT-I CTLs in the presence of FAS-Fc chimera protein (1 μM: $n = 42$, 10 μM: $n = 50$) or the control Fc protein ($n = 48$). Data are presented as mean and SD. **e** The viability of H2K$^b$OVA-myotube co-cultured with OT-I CTLs in the presence of FAS-Fc chimera protein (0.1 μM: $n = 57$, 1 μM: $n = 50$, 10 μM: $n = 44$) or the control Fc protein (0.1 μM: $n = 36$, 1 μM: $n = 33$, 10 μM: $n = 34$). Representative data of three independent experiments are shown. Log-rank test, followed by Holm–Sidak multiple comparisons. **$p < 0.01$.

H2K$^b$OVA-myotubes, the levels of HMGB1, IL-1α and IL-6 were increased in the co-culture of myotubes and OT-I CTLs (Fig. 4g–i). HMGB1 has different redox isoforms with distinct functions. The all-thiol HMGB1 and disulfide HMGB1 exert chemoattractant and cytokine activity, respectively[19]. The redox state of HMGB1 in the supernatants was analyzed by immunoblotting and utilizing the difference in electrophoretic mobility under nonreducing conditions[19]. While the all-thiol HMGB1 was detected as a single band with an apparent molecular weight of 28 kD both in reducing and nonreducing conditions, the disulfide HMGB1 was detected as a single band in nonreducing conditions with an apparent molecular weight of 26 kD but shifted in reducing conditions to 28 kD[19]. According to the electrophoretic pattern, HMGB1 in the supernatants of the co-culture of myotubes and OT-I CTLs was in the disulfide form (Supplementary Fig. 3). Inhibition of necroptosis in the myotubes by pretreating them with the inhibitor Nec1s prior to the co-culture suppressed the levels of these inflammatory mediators in the co-culture (Fig. 4g–i).

**Inhibition of necroptosis ameliorated muscle strength and inflammation in a murine model of PM.** Given the compelling in vitro data showing that CTL-induced muscle cell death and subsequent release of inflammatory mediators can be suppressed by necroptosis inhibition, we sought to determine whether necroptosis could be targeted in vivo using CIM, a murine model of PM. We first examined the muscle samples in CIM for the presence of apoptotic cells and for the expression of necroptosis-associated molecules. We found that none of the muscle fibers were TUNEL positive, while some of PAX7 positive satellite cells were stained with TUNEL (Fig. 5a, Supplementary Fig. 4a). Also, expression of RIPK1, RIPK3, MLKL, phosphorylated-S345 MLKL, FAS, CFLAR, and FAS was detected in the injured muscle fibers in CIM (Fig. 5b, Supplementary Fig. 4b, c), while these molecules were not observed in non-injured muscle fibers (Supplementary Fig. 4d). While necroptosis is morphologically characterized by cell swelling, some of the injured muscle fibers of CIM were rather shrunk, possibly representing atrophy

induced by inflammation[20,21] (Supplementary Fig. 4a, b). These observations were consistent with the histological findings of human PM.

In addition, we analyzed the effect of the deficiency of RIPK3 or MLKL in CIM. In both $Ripk3^{-/-}$ and $Mlkl^{-/-}$ CIM mice, the histological inflammation scores (Fig. 5c, e) as well as the necrotic areas (Fig. 5d, f) in the muscles were significantly decreased compared to those of the wild-type CIM mice, implying the involvement of necroptosis in the pathophysiology of CIM.

Next, we examined the therapeutic effect of necroptosis inhibition in CIM. CIM resulted in decrease of grip strength by about 16% on day 14 after immunization of C-protein (Fig. 5g). However, the decrease of grip strength was not observed in mice prophylactically treated with Nec1s from day 0. In addition, in mice treated with Nec1s on day 7 post immunization, when the CIM-induced decrease of grip strength and histological myositis were evident, we found that the grip strength was fully recovered on day 14 and 21. Histological analysis of the muscles revealed that the histological inflammation scores (Fig. 5h) as well as the necrotic areas (Fig. 5i) in the muscles decreased in mice treated prophylactically or therapeutically with Nec1s compared to those treated with a vehicle control. We also found that the proportion of TUNEL positive cells among PAX7 positive cells in CIM decreased in mice treated with Nec1s compared to those treated with the vehicle control (Fig. 5j). Consistent with the histological attenuation of muscle inflammation, the levels of IL-1α, IL-1β, and IL-6 in the muscles were lower in Nec1s-treated CIM mice than mice treated with the vehicle control (Supplementary Fig. 5a–c). These results indicate that inhibition of necroptosis not only suppressed CIM-induced muscle fiber death but also muscle inflammation and satellite cell death. Additionally, we examined the therapeutic effect of the inhibition of ferroptosis and pyroptosis, other types of regulated form of cell death with the necrotic morphological features[22]. Treatment with the inhibitor of ferroptosis, ferrostatin-1 (Fer-1)[23] or the inhibitors of pytoptosis, belnacasan (Vx765)[24] or disulfiram[25] did not ameliorate CIM-induced decrease of grip strength (Supplementary Fig. 6a), the histological inflammation scores (Supplementary

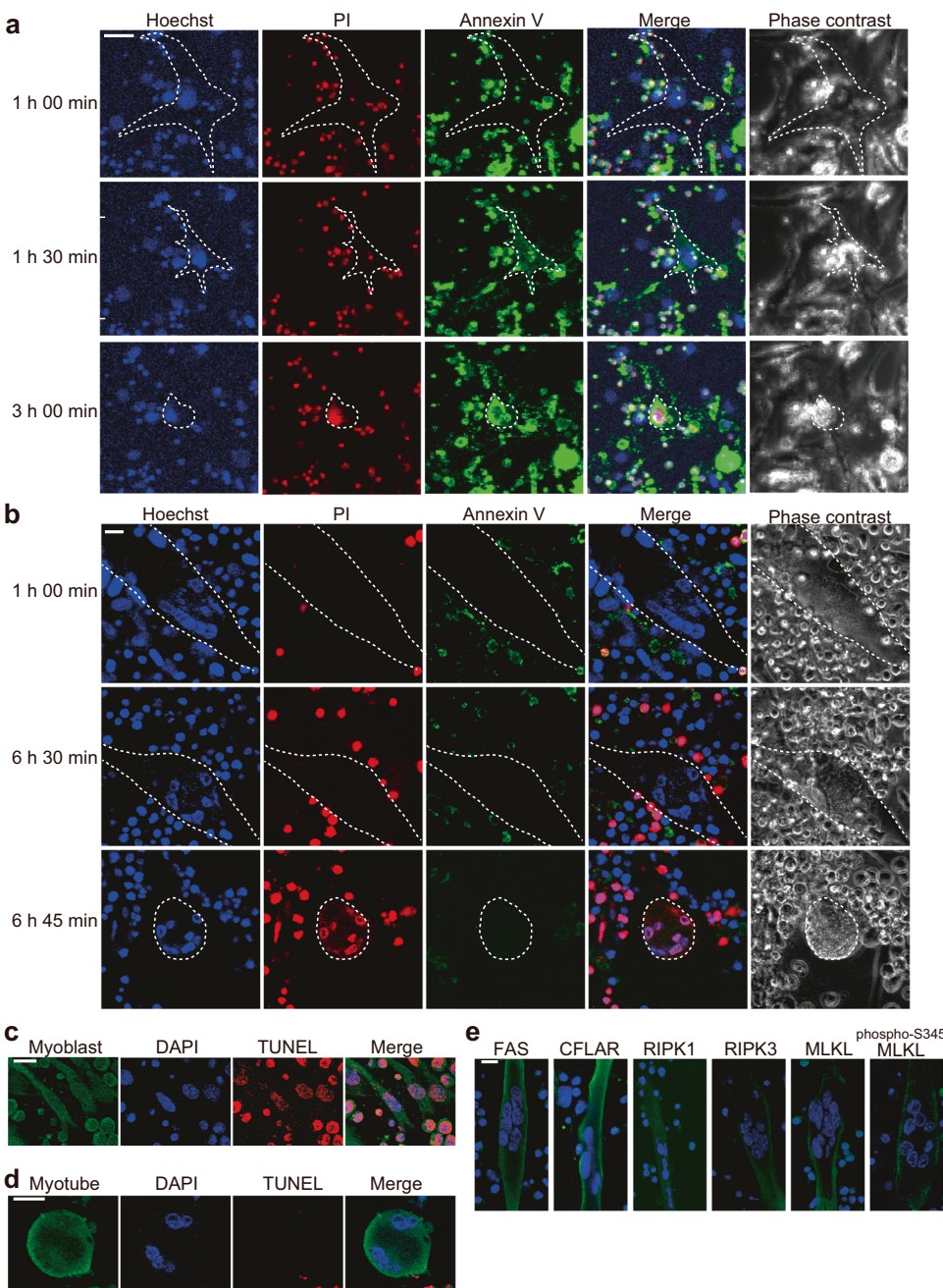

**Fig. 3 Different pattern of cell death between myoblasts and myotubes in the co-culture with CTLs. a, b** The representative confocal z-stack images of time-lapse analysis of H2K$^b$OVA-myoblasts (**a**) or H2K$^b$OVA-myotubes (**b**) co-cultured with OT-I CTLs in the presence of Annexin V (green) and PI (red). Nuclei of the cells were stained with Hoechst 33342 (blue). Images were taken at the indicated times from the starting of the co-culture. The cells surrounded by dotted line indicate dying myoblast and myotube. Scale bars indicate 20 μm. Representative data of three independent experiments are shown. **c, d** The TUNEL staining (red) of H2K$^b$OVA-myoblasts (**c**) or H2K$^b$OVA-myotubes (**d**) co-cultured with OT-I CTLs for 4 or 16 h, respectively. The myoblasts and myotubes were pre-labelled with CellTracker (green). Confocal z-stack images are shown. Scale bars indicate 20 μm. Representative data of two independent experiments are shown. **e** The immunofluorescence staining of FAS, CFLAR, RIPK1, RIPK3, MLKL, and phosphorylated MLKL at S345 (phospho-S345 MLKL; green) in H2K$^b$OVA-myotubes co-cultured with OT-I CTLs for 8 h. Nuclei were counterstained with DAPI (blue). Scale bar indicates 10 μm. Representative data of three independent experiments are shown.

Fig. 6b), or the necrotic areas (Supplementary Fig. 6c) in the muscles, suggesting these cell death pathways were less involved in the pathogenesis of CIM.

**HMGB1 mediates muscle inflammation.** We hypothesized that the therapeutic effect of necroptosis inhibition on muscle inflammation was mediated by reducing the release of DAMPs from dying muscle fibers. We found that the level of HMGB1 in

the serum was markedly increased in CIM (Fig. 6a). Strikingly, treatment with the necroptosis inhibitor Nec1s suppressed CIM-induced increase of HMGB1 by about 90%. Consistent with a previous study of human PM[26], HMGB1 was highly expressed in the muscle fibers in the areas where inflammatory infiltrates were observed, indicating that the muscle fibers are a dominant source of HMGB1 in CIM (Fig. 6b). Next, we investigated the redox state of HMGB1 by immunoprecipitating HMGB1 from the

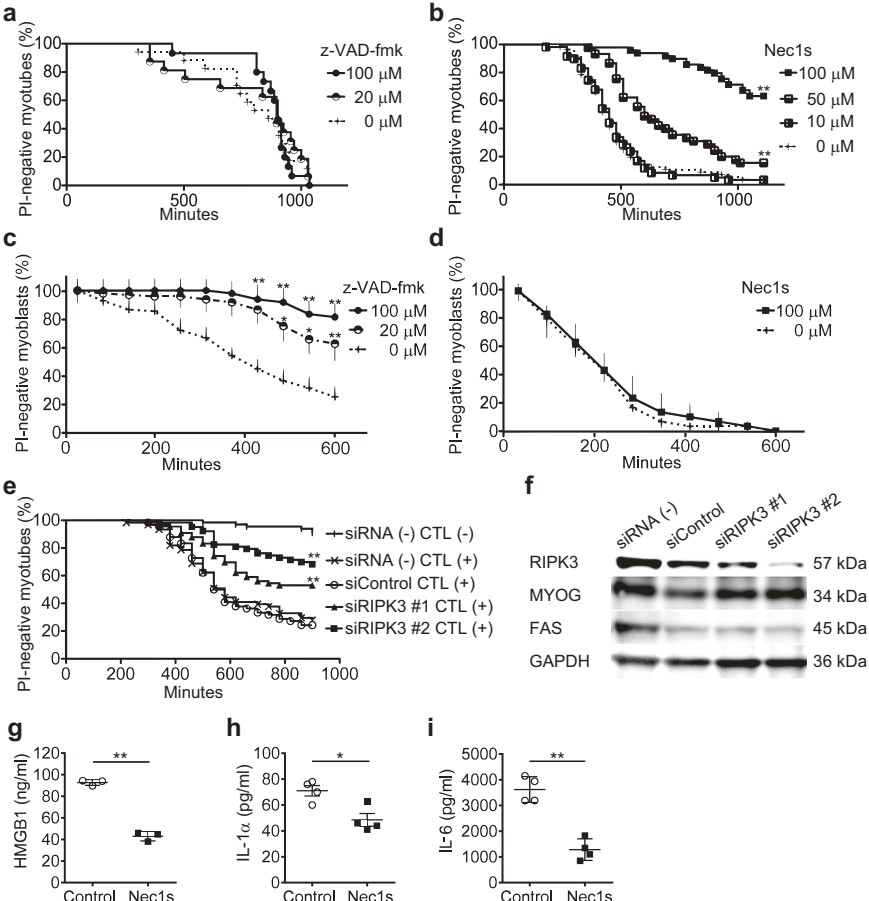

**Fig. 4 Effect of apoptosis or necroptosis inhibition on CTL-induced cell death in myotubes and myoblasts. a**, **b** The viability of z-VAD-fmk- or Nec1s-pretreated H2K$^b$OVA-myotubes in the co-culture with OT-I CTLs. z-VAD-fmk 0 μM: $n = 30$, 20 μM: $n = 30$, 100 μM: $n = 30$. Nec1s 0 μM: $n = 56$, 10 μM: $n = 55$, 50 μM: $n = 46$, 100 μM: $n = 50$. Log-rank test, followed by Holm–Sidak multiple comparisons. **$p < 0.01$. **c**, **d** The viability of z-VAD-fmk- or Nec1s-pretreated H2K$^b$OVA-myoblasts in the co-culture with OT-I CTLs. z-VAD-fmk 0 μM: $n = 50$, 20 μM: $n = 48$, 100 μM: $n = 59$. Nec1s 0 μM: $n = 33$, 100 μM: $n = 30$. Data are presented as mean and SD. Two-way ANOVA test, followed by Dunnett's test. *$p < 0.05$, **$p < 0.01$. **e** The viability of H2K$^b$OVA-myotubes transfected with scrambled siRNA (siControl: $n = 66$), siRNA specific for *Ripk3* (siRIPK3 #1: $n = 67$, siRIPK3 #2: $n = 63$), or without transfection (siRNA (−): $n = 61$) in the co-culture with OT-I CTLs. Log-rank test, followed by Holm–Sidak multiple comparisons. **$p < 0.01$. **a–e** The data represent three independent experiments. **f** The protein expression of RIPK3, MYOG, FAS, and GAPDH evaluated with western blotting in H2K$^b$OVA-myotubes transfected with siControl or siRIPK3. The data represent two independent experiments. **g–i** The levels of HMGB1 (**g**), IL-1α (**h**), and IL-6 (**i**) in the co-culture of Nec1s-pretreated or untreated H2K$^b$OVA-myotubes and OT-I CTLs for 20 h. IL-1β was below detectable level in the culture supernatants of all conditions. The levels in the mono-culture of OT-I CTLs are subtracted from those in the co-culture. Data are presented as mean and SD of triplicate (**g**) or quadruplicate (**h**, **i**) experiments. Student's t-test. *$p < 0.05$, **$p < 0.01$.

muscle homogenates and the serum of CIM. Majority of HMGB1 in both the muscle homogenates and the serum of CIM was in the all-thiol form and the levels of both the all-thiol and disulfide form was increased in the samples of the CIM mice compared to mice without CIM. Of note, the disulfide form of HMGB1 was barely observed in the samples of the mice without CIM (Supplementary Fig. 7a, b). To confirm if the increased release of HMGB1 is involved in the muscle inflammation, CIM mice were treated with anti-HMGB1 antibodies[27]. Anti-HMGB1 antibody-treated mice exhibited significantly less muscle inflammation (Fig. 6c) and necrotic areas in the muscles (Fig. 6d) as well as more grip strength (Fig. 6e) compared to mice treated with isotype control antibodies. These results support our hypothesis that HMGB1 contributes to PM and blockade of HMGB1 has similar effects as inhibiting necroptosis with Nec1s in CIM.

## Discussion

In this study, we found that the pattern of cell death in muscle fibers is necroptosis using an integrative analysis including

histological imaging of human muscle biopsy samples from PM/DM patients and functional studies with models of PM in vitro and in vivo. Dying muscle fibers release high levels of HMGB1 leading to further acceleration of muscle inflammation and subsequent muscle injury in PM. Our observations indicate that muscle cells are not merely passive responders targeted by CTLs, but rather are aggressors that actively promote muscle inflammation in PM.

Since CTL-induced necroptosis of myotubes is FAS-FASLG dependent, inhibition of FAS-signaling in muscle fibers could be a therapeutic strategy in PM. However, systemic blockade of FAS-FASLG interaction may exacerbate PM. Interaction of FAS and FASLG plays a crucial role in activation-induced cell death of T cells[28], which is essential for eliminating autoreactive T cells. In fact, impairment of FAS-mediated apoptosis by mutations in *FAS* gene causes autoimmune lymphoproliferative syndrome characterized by chronic lymphoproliferation and autoimmune manifestations[29,30]. Therefore, viable therapeutic strategies for PM need to specifically target muscle cell death and at the same time avoid other immune side effects. Our

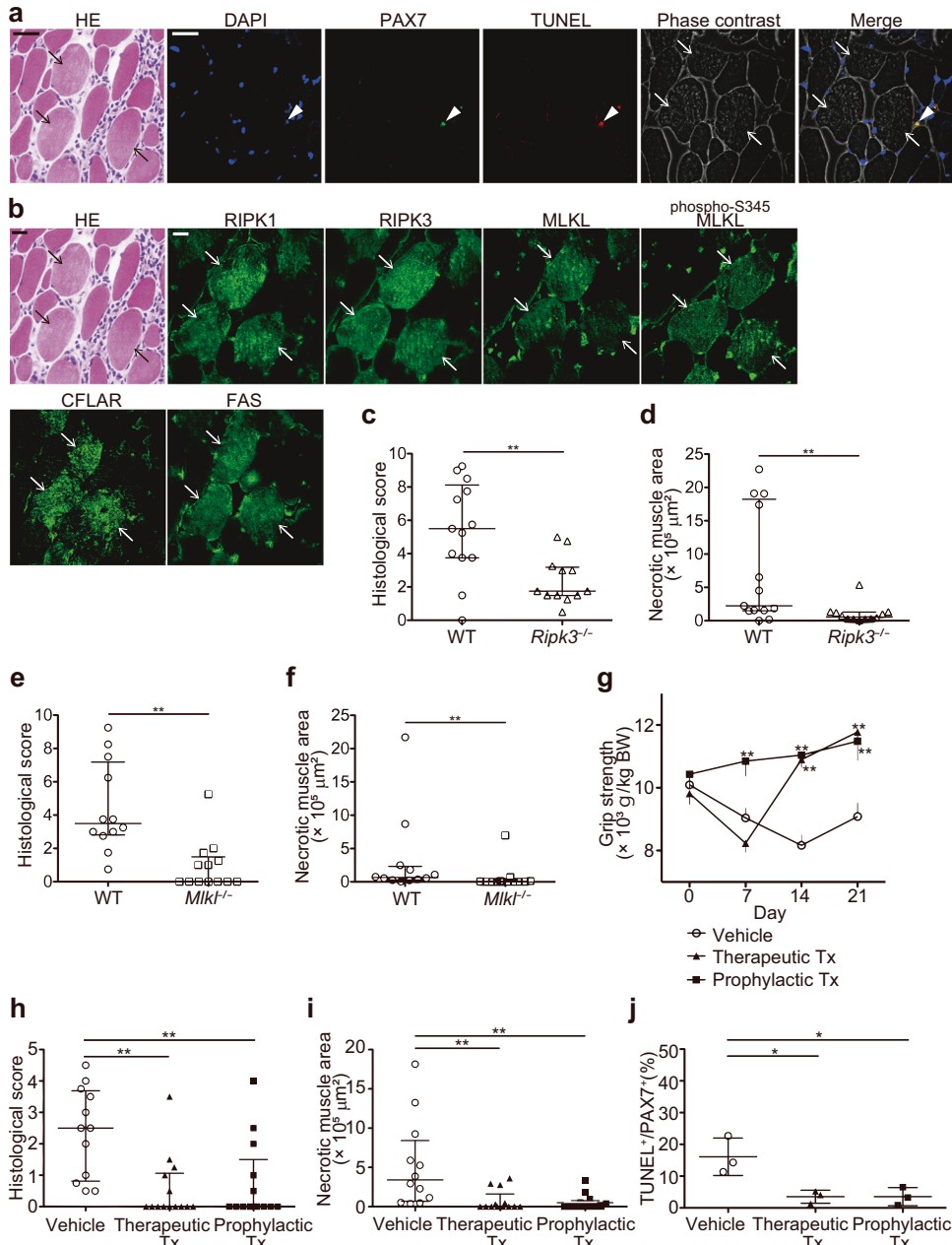

**Fig. 5 The involvement of necroptosis in the muscle inflammation and muscle weakness in CIM. a** HE, immunofluorescence staining against PAX7 (green), and the TUNEL staining (red). Nuclei were counterstained with DAPI (blue). The arrowhead indicates PAX7 positive TUNEL positive satellite cell. **b** HE and immunofluorescence staining (green) for the expression of RIPK1, RIPK3, MLKL, phosphorylated-S345 (phospho-S345) MLKL, CFLAR, and FAS in the muscle specimens of CIM mice. **a, b** The arrows indicate the dying muscle fibers. Scale bars indicate 20 μm. Each histological examination was performed three times and the representative images are shown. **c, d** The histological scores (**c**) of the severity of myositis and the area of necrotic muscle fibers (**d**) in wild-type (WT, $n = 13$) or $Ripk3^{-/-}$ ($n = 12$) mice on day 14 of CIM. **e, f** The histological scores (**e**) and the area of necrotic muscle fibers (**f**) in WT ($n = 12$) or $MLKL^{-/-}$ ($n = 13$) mice on day 14 of CIM. **c–f** Data are presented as median ± interquartile range. Mann–Whitney $U$ test. **$p < 0.01$. **g** The grip strength of CIM mice treated with Nec1s or the vehicle ($n = 12$). The treatment was started immediately after the immunization in the prophylactic group (Prophylactic Tx, $n = 13$), or from day 7 in the therapeutic group (Therapeutic Tx, $n = 14$). BW; Body weight of the mice. Data are presented as mean ± SD. Two-way ANOVA test, followed by Dunnett's multiple comparison test. **$p < 0.01$. **h, i** The histological scores (**h**) and the area of necrotic muscle fibers (**i**) on day 21 of CIM. Vehicle; $n = 12$, Therapeutic Tx; $n = 14$, Prophylactic Tx; $n = 13$. Data are presented as median ± interquartile range. Kruskal–Wallis test, followed by Dunn's test. **$p < 0.01$. **j** Proportion of TUNEL$^+$ cells out of PAX7$^+$ cells in the muscles of CIM on day 21. Vehicle; $n = 3$, Therapeutic Tx; $n = 3$, Prophylactic Tx; $n = 3$. Data are presented as mean ± SD. One-way ANOVA test, followed by Dunnett's test. *$p < 0.05$. **c–j** Data represent two independent experiments.

study suggests that inhibition of necroptosis or HMGB1 would be a promising alternative strategy to suppress CTL-induced muscle injury without excessive activation of autoreactive T cells.

The systemic inhibition of necroptosis in vivo could also affect the cell death of immune cells. While the pattern of cell death in most of the immune cells appears to be apoptosis based on our observations using the in vitro model of CTL-induced muscle cell

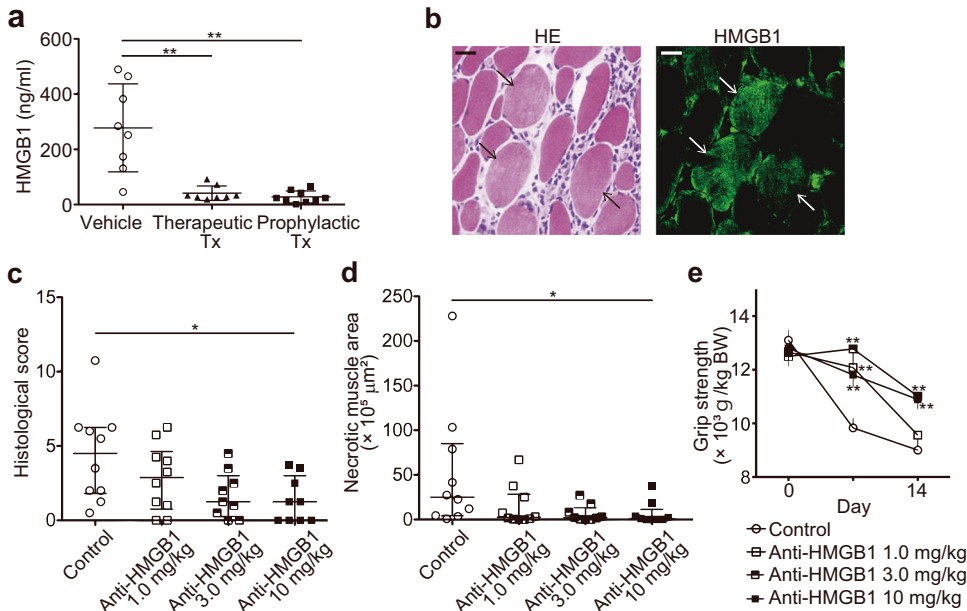

**Fig. 6 The involvement of HMGB1 in the muscle inflammation and muscle weakness in CIM. a** The serum levels of HMGB1 in CIM mice on day 21. The data are presented as mean ± SD. One-way ANOVA test, followed by Dunnett's test. **$p < 0.01$. **b** HE and immunofluorescence staining of the muscle specimens of CIM for the expression of HMGB1. The arrows indicate the dying muscle fibers. Scale bars indicate 20 μm. **c** The histological scores on day 14 in CIM mice treated with 1.0 mg/kg ($n = 10$), 3.0 mg/kg ($n = 9$), or 10 mg/kg ($n = 9$) of anti-HMGB1 antibodies (anti-HMGB1) or 10 mg/kg of the control antibodies (Control: $n = 10$). The data are presented as median ± interquartile. Kruskal–Wallis test, followed by Dunn's test. *$p < 0.05$. **d** The areas of necrotic muscle fibers on day 14 of CIM. Data are presented as median ± interquartile range. Kruskal–Wallis test, followed by Dunn's test. *$p < 0.05$. **e** The grip strength of CIM mice treated with anti-HMGB1 antibodies or the control antibodies. Data are presented as mean ± SD. Two-way ANOVA test, followed by Dunnett's test. **$p < 0.01$. **c**–**e** Data represent two independent experiments.

death and the histological analysis of PM/DM and CIM, necroptosis-associated molecules are expressed in some immune cells in the muscle specimens of PM/DM and CIM. Necroptosis reduced proliferative capacity of antigen-specific CTLs in chronic progressor of HIV[31] and other types of immune cells such as macrophages could undergo necroptosis in certain experimental conditions[32]. Our study revealed that treatment with Nec1s ameliorated CTL-induced muscle cell death as well as muscle inflammation in CIM, indicating that necroptosis is not a dominant form of cell death of immune cells or even if some of the infiltrating immune cells undergo necroptosis, its inhibition might have been protective in the pathogenesis in CIM.

Marked increase in the levels of HMGB1 in both serum and muscle fibers, and the therapeutic effect of anti-HMGB1 antibodies in CIM implicate an important role of HMGB1 produced by the muscle cells in the pathogenesis of CIM. HMGB1 acts as an innate adjuvant to promote the production of inflammatory cytokines and antigen presentation in myeloid cells[33], and it has been reported to be involved in a variety of pathological conditions including rheumatoid arthritis[34], infections[35], and brain injuries[27]. In patients with inflammatory myopathies, the serum levels of HMGB1 were elevated, especially in the patients with interstitial lung diseases, which is a major complication leading to fatal outcomes[36]. Although the role of HMGB1 in the pathogenesis of interstitial lung diseases has not been elucidated, inhibition of necroptosis might also be beneficial for interstitial lung diseases complicated with inflammatory myopathies.

We observed quick improvement of CIM-induced muscle weakness by Nec1s treatment. This result suggests the improvement of muscle fiber functions in addition to the inhibition of the necroptosis by Nec1s. HMGB1 has been reported to induce dysfunction of muscle fibers via its receptor TLR4[37]. HMGB1 triggers TLR4 and RIPK3-dependent necroptosis of hepatocytes in acetaminophen-induced liver injury[38]. Preventing HMGB1

release from injured muscle fibers is probably one of the key mechanisms by which necroptosis inhibition attenuates CIM-induced muscle weakness.

In the treatment of PM, it is important to promote muscle regeneration to help recover from muscle weakness in addition to the suppression of muscle injuries by CTLs. While HMGB1 is a crucial inflammatory mediator in the pathogenesis of PM, it also acts as a potent regenerative factor of muscles[39]. Therefore, the long-term blockade of HMGB1 in the chronic phase of the disease might interfere with the recovery of muscle strength in PM patients. Considering the therapeutic effect of necroptosis inhibition to suppress muscle fiber death and subsequent release of HMGB1, inhibition of necroptosis may provide a good therapeutic option in the acute phase of the disease when muscle cells are injured by CTLs.

Unexpectedly, the treatment with Nec1s in vivo suppressed not only necroptosis of muscle fibers but also apoptosis of satellite cells, which is another benefit of necroptosis inhibition that we found in this study. The CTL-induced apoptosis of satellite cells depends on GZMB. GZMB-mediated cytotoxicity of CTLs was promoted by inflammatory cytokines including IL-1, IL-6, IL-2, IL-15, and type I interferons[40–43]. In addition to the crucial role of RIPK3 in necroptosis, RIPK3 has been reported to promote the production of inflammatory cytokines via NF-κB that is independent of necroptosis[44]. Indeed, mice lacking RIPK3 or MLKL were strikingly resistant to CIM-induced muscle damage. The decrease of multiple inflammatory cytokines by both the inhibition of necroptosis of muscle fibers and NF-κB pathway might be responsible for suppression of the CTL-induced apoptosis of satellite cells by Nec1s in CIM.

Resistance against apoptosis has been reported in different types of terminally differentiated cells including cardiomyocytes, neurons, keratinocytes, and pancreatic acinar cells as well as muscle fibers[45–48]. Similar to these cells, muscle fibers also

express antiapoptotic molecules such as CFLAR[8], BCL2[49] and XIAP[50], and lose the expression of pro-apoptotic molecules such as APAF1 during the differentiation from the progenitor cells[51]. While resistance to apoptosis in the terminally differentiated cells is crucial to maintain the tissue function and structure of the organs, the machinery of the resistance could provoke tissue damage in pathogenic conditions such as myocarditis, neurodegenerative disease, psoriasis, and chronic pancreatitis as well as inflammatory myopathy via necroptosis[45–48]. Several necroptosis inhibitors have been tested in the clinical trials for treatment of psoriasis, ulcerative colitis, and neurodegenerative diseases[52,53]. In a phase 2 multicenter randomized placebo-controlled study for psoriasis patients, treatment with a RIPK1 inhibitor improved the severity of plaque lesions compared with placebo without any severe drug-related adverse effects[52]. Clinical trials to test the therapeutic and side effects of necroptosis inhibition are awaited in PM, given the promising therapeutic effects in our pre-clinical studies.

Inflammatory myopathies are the systemic diseases with a variety of extramuscular involvements, including the skin, lung, heart, joint, and gastro-intestinal tract[9]. Since there are no immunological animal models of inflammatory myopathies reproducing these extramuscular involvements[54], the effect of inhibition of necroptosis on these organs could not be evaluated in vivo. Given the involvement of necroptosis in inflammatory diseases of various organs and the efficacy of necroptosis inhibitions in corresponding animal models such as dermatitis[47,52], acute respiratory distress syndrome[55,56], cardiomyositis[45], arthritis[57], and colitis[58], systemic inhibition of necroptosis could also be a potential therapy for the extramuscular involvement of inflammatory myopathies.

While the physiological relevance of necroptosis remains largely unclear, necroptosis is indicated to be involved the inflammatory response in infectious diseases[31] and tumor immunity[59]. For the clinical application, it is necessary to take these potential side effects of long-term inhibition of necroptosis into consideration.

In conclusion, inhibition of necroptosis in muscle fibers would be a novel therapeutic strategy to ameliorate muscle weakness via suppression of muscle cell death and inflammation in PM. Since this type of muscle cell-directed therapy does not directly suppress immune cells or inflammatory mediators, it is a promising alternative to current immunosuppressive therapies for PM with potentially less infectious complications.

## Methods

**Patients and muscle biopsy**. Muscle specimens were obtained using percutaneous conchotome muscle biopsy technique[60] from the tibialis anterior muscle of twelve untreated adult patients with PM ($n = 9$) or DM ($n = 3$) who met the Bohan and Peter criteria[61] and 2017 European League Against Rheumatism/American College of Rheumatology (EULAR/ACR) classification criteria for adult and juvenile idiopathic inflammatory myopathies[62] at Department of Rheumatology, Tokyo Medical and Dental University (TMDU) between October 2016 and April 2020. The median (interquartile range) age of the patients was 35 (34, 61) and 50 (25, 59) in PM and DM, respectively. Five and one female patients were included in PM and DM, respectively. The conchotome muscle biopsy from the tibialis anterior muscle is a simple and less invasive procedure with a low complication rate, comparing to conventional muscle biopsy. Its validity for the histopathological evaluation of PM/DM has been well evaluated as the samples obtained from the tibialis anterior muscle of PM/DM patients with the procedure showed no significant difference in the frequency of inflammatory infiltrates, necrotic muscle fibers, and regenerating muscle fibers compared to those obtained from the proximal muscles of the same patients[21]. All the patients included in the analysis showed necrotic muscle fibers, which were identified with reduced eosin staining in the cytoplasm but did not have the characteristic feature of immune-mediated necrotizing myopathy[63]. The patients who were suspicious of cancer-associated, viral, or immune check point inhibitor-associated myositis were excluded. The clinical, serological, and histopathological features of the patients were shown in Supplementary data 1. The study protocols were approved by the institutional review board at TMDU and are in accordance with the principles of the Declaration of Helsinki. Written informed consent was obtained from all participants.

**Antibodies and reagents**. The following antibodies and reagents were used for immunofluorescence staining: anti-PAX7 (PAX7497, Abcam, Cambridge, UK, 1:200), anti-RIPK1 (rabbit polyclonal, NOVUS Biologicals, Litleton, Colorado, USA, #NBP1-77077, 1:200), anti-RIPK3 (rabbit polyclonal, Abcam, Cambridge, UK, #ab152130, 1:100), anti-MLKL (rabbit polyclonal, LSBio, Seattle, Washington, USA, #LS-C334151, 1:100), anti-MLKL (phospho S358, EPR9514, Abcam, Cambridge, UK, 1:250), anti-MLKL (phospho S345, EPR9515(2), Abcam, Cambridge, UK, 1:100), anti-CASP8 (Full length, rabbit polyclonal, Abcam, Cambridge, UK, #ab4052, 1:100), anti-CASP8 (active form p18 subunit, 2B12.1, Merck, Kenilworth, New Jersey, USA, 1:250), anti-FAS (rabbit polyclonal, Abcam, Cambridge, UK, #ab82419, 1:100), anti-CFLAR (rabbit polyclonal, Abcam, Cambridge, UK, #ab8421, 1:100), anti-HMGB1 (rabbit polyclonal, Abcam, Cambridge, UK, #ab18256, 1:500), polyclonal rabbit IgG (Abcam, Cambridge, UK, #ab37415), monoclonal rabbit IgG (EPR25A, Abcam, Cambridge, UK), monoclonal mouse IgM (GC323, Merck, Kenilworth, New Jersey, USA), anti-mouse IgG-Alexa Fluor 647 (Thermo Fisher Scientific, Waltham, Massachusetts, USA, #A21237), anti-rabbit IgG-Alexa Fluor 647 (Thermo Fisher Scientific, Waltham, Massachusetts, USA, #A21245), anti-mouse IgM-Alexa Fluor 594 (Thermo Fisher Scientific, Waltham, Massachusetts, USA, #A21044) antibodies, and Fluoromount-G Mounting Medium with DAPI (Thermo Fisher Scientific, Waltham, Massachusetts, USA). For the immunohistochemical staining, anti-MLKL (phospho S358, EPR9514, Abcam, Cambridge, UK, 1:250), anti-CD8 (C8/144B, Nichirei, Tokyo, Japan, 1:200), anti-CD4 (1F6, Nichirei, Tokyo, Japan, 1:100), anti-CD68 (PG-M1, Dako Cytomation, Glostrup, Denmark, 1:100), anti-CD20cy (L26, Dako Cytomation, Glostrup, Denmark, 1:100), anti-C5b-9 (aE11, Dako Cytomation, Glostrup, Denmark, 1:100), anti-HLA-ABC (W6/32, Dako Cytomation, Glostrup, Denmark, 1:500), monoclonal rabbit IgG (EPR25A, Abcam, Cambridge, UK), mouse IgG1 (MAB002 R&D Systems, Minneapolis, Minnesota, USA), peroxidase-labeled amino acid polymer-conjugated goat anti-rabbit IgG (Dako Cytomation, Glostrup, Denmark), anti-mouse IgG (Dako Cytomation, Glostrup, Denmark), and diaminobenzidine (Dako Cytomation, Glostrup, Denmark). For the detection of apoptosis in human and mouse muscle specimens and cultured cells, Click-iT TUNEL Alexa Fluor 594 Imaging Assay kit (Invitrogen, Carlsbad, California, USA) was used. For the time-lapse imaging, the following reagents were used: CellTracker™ Green (Invitrogen, Carlsbad, California, USA), benzyloxycarbonyl-Val-Ala-Asp-fluoromethylketone (z-VAD-fmk, BACHEM, Bubendorf, Switzerland), necrostatin-1s (Nec1s, Haoyuan ChemExpress Co., Ltd., Shanghai, China), recombinant mouse FAS-Fc chimera protein (R&D Systems, Minneapolis, Minnesota, USA), recombinant human Fc protein (R&D Systems, Minneapolis, Minnesota, USA), Hoechst 33342 (Thermo Fisher Scientific, Waltham, Massachusetts, USA), propidium iodide (PI; Invitrogen, Carlsbad, California, USA), and FITC-conjugated Annexin V (BioLegend, San Diego, California, USA). For the western blotting, the following regents were used: anti-RIPK3 (rabbit polyclonal, Abcam, Cambridge, UK, #ab152130, 1:1000), anti-MYOG (EPR4789, Abcam, Cambridge, UK, 1:1000), anti-FAS (rabbit polyclonal, Abcam, Cambridge, UK, #ab82419, 1:1000), anti-GAPDH (D4C6R, Cell Signaling Technology, Danvers, Massachusetts, USA, 1:1000), anti-HMGB1 (rabbit polyclonal, Abcam, Cambridge, UK, #ab18256, 1:1000), horseradish peroxidase (HRP)-conjugated goat anti-rabbit IgG, goat anti-mouse IgG antibodies (Cell Signaling Technology, Danvers, Massachusetts, USA), and Clean-Blot IP detection Reagent (Thermo Fisher Scientific, Waltham, Massachusetts, USA).

**Manual muscle testing**. Total manual muscle testing score of 8 muscles was evaluated before the initiation of treatment based on Kendall's 0–10 point scale in neck extensor, deltoid, biceps, gluteus maximus, iliopsoas, quadriceps, wrist extensor, and ankle dorsiflexor muscles with a maximal score of 150. All of the muscles, except neck extensor muscle, were bilaterally evaluated[64,65].

**Measurement of serum CK and autoantibodies**. Serum samples were obtained from untreated donors of PM and DM patients who had muscle biopsy, and were analyzed for the levels of CK and autoantibodies in the clinical laboratory. The CK levels were measured by a standard enzymatic method. ANA was measured by a fluorescent antibody method. Anti-ARS antibodies, which bind to Jo-1, PL-7, PL-12, EJ or KS, were measured by enzyme-linked immunosorbent assay (ELISA). Anti-Jo-1, anti-SS-A, and anti-SS-B antibodies were measured by ouchterlony method or ELISA. Anti-TIF1γ, anti-Mi-2, anti-melanoma differentiation associated gene 5 (MDA5), anti-RNP, and anti-centromere antibodies were measured by an enzyme immunoassay. Anti-mitochondria M2 antibodies were measured by a chemiluminescent enzyme immunoassay.

**Mice**. OVA-specific class I restricted T cell receptor transgenic mice (OT-I) and C57BL/6 mice were purchased from Charles River Japan (Kanagawa, Japan). The splenocytes from mutant OT-I mice lacking PRF1 or GZMB[66] were provided by Dr. Naoko Okiyama (Tsukuba University, Ibaraki, Japan). C57BL/6 $Ripk3^{-/-}$ mice and C57BL/6 $Mlkl^{-/-}$ mice were described previously[67,68]. The mice were housed at a standard temperature ($21 \pm 1$°C) under a 12-h light: 12-h dark cycle in a

humidity-controlled (55 ± 5%) environment with ad libitum access to standard diet (CE-2, CLEA) and water. Mice were bred and experimental procedures were carried out at the center for animal research in TMDU and The Walter and Eliza Hall Institute of Medical Research (WEHI). All animal experiments were approved by the Institutional Animal Care and Use Committee of TMDU and Animal Ethics Committee in WEHI and were performed in accordance with the guidelines of both institutes and both countries.

**Preparation of OT-I CTLs**. OT-I CTLs were prepared as described previously[15]. Briefly, the splenocytes of male or female OT-I mice at the age of 2–4 months were stimulated with the SIINFEKL peptide (BACHEM, Bubendorf, Switzerland) in Roswell Park Memorial Institute (RPMI)-1640 medium (Sigma-Aldrich, St. Louis, Montana, USA) supplemented with 10% fetal bovine serum (FBS), penicillin, and streptomycin for 3 days. CD8$^+$ T cells were purified using magnetic beads (Miltenyi Biotech, Bergisch Gladbach, Germany) according to the manufacturer's protocol. The purity of CD8$^+$ T cells was greater than 98%.

**Cell culture of H2K$^b$OVA-myoblasts and H2K$^b$OVA-myotubes**. C2C12 cells were stably transduced with mouse β2-microglobulin, SIINFEKL peptide derived from ovalbumin, and H2K$^b$ as described previously[15]. H2K$^b$OVA-myoblasts were cultured in high-glucose Dulbecco's modified Eagle's medium (DMEM; Sigma-Aldrich, St. Louis, Montana, USA) supplemented with 10% FBS, penicillin, and streptomycin. To differentiate the myoblasts to myotubes, the myoblasts were cultured until reaching confluence, and then cultured in DMEM supplemented with 2% horse serum for 4–5 days.

**siRNA silencing**. H2K$^b$OVA-myoblasts were plated at $5 × 10^3$ cells/cm$^2$ in DMEM containing 10% FBS. After 48 h, the cells were transfected with siRNAs or scrambled siRNA (Thermo Fisher Scientific, Waltham, Massachusetts, USA) at 10 nM using Lipofectamine RNAiMAX Reagent (Thermo Fisher Scientific, Waltham, Massachusetts, USA) in DMEM containing 2% horse serum. After 24 h, the culture media was replaced with DMEM containing 2% horse serum to differentiate the transfected cells into H2K$^b$OVA-myotubes.

**Time-lapse imaging**. H2K$^b$OVA-myoblasts or H2K$^b$OVA-myotubes were co-cultured with OT-I CTLs as described previously[15]. Briefly, H2KbOVA myoblasts were plated on 35-mm glass-bottomed dishes (Iwaki Glass, Tokyo, Japan) or eight-well chambered slides (Thermo Fisher Scientific, Waltham, Massachusetts, USA) 1 day or 6–7 days before the coculture for the analysis of myoblasts or myotubes, respectively. H2KbOVA-myoblasts or myotubes were labeled with CellTracker$^{TM}$ Green according to the manufacturer's instructions, as needed. H2K$^b$OVA-myoblasts or myotubes were co-cultured with 10$^6$ cells/cm$^2$ OT-I CTLs in DMEM supplemented with 10% FBS in 5% CO$_2$ (v/v) at 37 °C. In some experiments, the myoblasts or myotubes were pre-treated with z-VAD-fmk or Nec1s for 24 h before the co-culture. Images were acquired with a confocal microscope FV10i-W (Olympus, Tokyo, Japan) every 10–30 min for the indicated time, and processed with FV10-ASW and ImageJ softwares. All of the myoblasts or myotubes in five to eight randomly taken microscopic fields which contain about 5–10 myoblasts or 5–20 myotubes per field were evaluated.

**CIM induction and the treatment of mice**. Female C57BL/6 mice at the age of 8 weeks and female and male C57BL/6 Ripk3$^{−/−}$ mice, C57BL/6 Mlkl$^{−/−}$ mice, or their littermates at the age of 6–10 weeks were immunized intradermally with 200 μg of human C protein fragments emulsified in Complete Freund's Adjuvant (CFA) containing 100 μg of heat-killed Mycobacterium butyricum (Difco, Franklin Lakes, New Jersey, USA)[69]. The emulsified C protein fragments were injected at 4 sites: the bilateral base of the tail and the base of bilateral hind limbs of the mice. The emulsified CFA without C protein was also injected subcutaneously into the base of bilateral forelimbs. At the same time, 250 ng of pertussis toxin (Seikagaku Kogyo, Tokyo, Japan) in 200 μl of 0.02% Triton X-100 (Sigma, St. Louis, Montana, USA) was intraperitoneally injected (IP). For the treatment of CIM with Nec1s, Nec1s was dissolved in dimethyl sulfoxide (DMSO, 2% w/v), and then diluted with 20% polyethylene glycol (PEG) 300 (ALDRICH, St. Louis, Montana, USA) solution at a final concentration of 1 mg/ml. Nec1s was administered via IP at 20 mg/kg. The treatment was started immediately after the immunization in the prophylactic group, or from day 7 in the therapeutic group. The injection was repeated every 12 h for 21 days in the prophylactic group, or for 14 days in the therapeutic group. In the control group, DMSO in PEG solution was administered as vehicle. The muscle specimens were collected on day 21 of CIM for histological evaluation except where otherwise noted. For the treatment of CIM with Fer-1 (Selleck Chemicals, Houston, Texas, USA), Vx765 (Selleck Chemicals, Houston, Texas, USA), and disulfiram (MedChemExpress, Monmouth Junction, New Jersey, USA), the compounds were dissolved in DMSO (2% w/v), and then diluted with 20% PEG 300 solution at a final concentration of 1 mg/ml, 5 mg/ml, and 5 mg/ml, respectively. Fer-1, Vx765, or disulfiram was administered via IP at 10, 50, and 50 mg/kg, respectively. The treatment was started immediately after the immunization and the injection was repeated every day for 21 days. In the control group, DMSO in PEG solution was administered as vehicle. The muscle specimens were collected on day 21 of CIM for histological evaluation. Anti-HMGB1 antibodies (#10–22,

produced by Okayama University, Okayama, Japan[27]) were administered every other days via IP at 1.0, 3.0, or 10 mg/kg. The class-matched antibodies (rat anti-keyhole limpet hemocyanin monoclonal antibodies, produced by Okayama University, Okayama, Japan) were used as the control. The muscle specimens were collected on day 14 of CIM for histological evaluation.

**Assessment of muscle strength of CIM**. The muscle strength of the mice was assessed using a grip strength meter for mice (Muromachi, Tokyo, Japan) as described previously[70]. The mice were allowed to grasp the grid of the strength meter with four limbs. The tail of the mice was gently pulled backward until its grasp was broken. The peak force was determined as the grip strength. Five consecutive measurements were made within 30 s. The measurements were performed between 1 p.m. and 3 p.m. once a week. The average of three measurements excluding the maximum and the minimum values was determined as the muscle strength. The muscle strength was normalized by dividing the values by body weight.

**Histological evaluation of CIM**. The hematoxylin and eosin (HE) stained 10 μm sections of the quadriceps and hamstrings were examined in a blinded manner for the presence of mononuclear cell infiltration and degeneration of the muscle fibers. The severity of myositis was graded histologically on the scales of 1–6, where 1 = involvement of 1 muscle fiber, 2 = involvement of 2–5 muscle fibers, 3 = involvement of 6–15 muscle fibers, 4 = involvement of 16–30 muscle fibers, 5 = involvement of 31–100 fibers, 6 = involvement of >100 muscle fibers. When multiple lesions with the same grade were found, 0.5 was added to the grade. The score of each muscle was the average grade score of 2 different sections. The histological scores of the individual mice were sum scores of the quadriceps and hamstrings[71]. The necrotic area of the muscle was measured using ImageJ software.

**Immunofluorescence staining**. Muscle specimens from human or mice were frozen for histological analysis. Sections were made in 10 μm thickness. The slides were fixed with 4% paraformaldehyde, permeabilized with 0.1% (v/v) Triton X-100 as needed, blocked with 10% goat serum (Dako Cytomation, Glostrup, Denmark) and 0.1 M glycine in PBS, and then incubated with primary antibodies followed by secondary antibodies. Images were obtained with a confocal laser scanning microscope FV10i-DOC (Olympus, Tokyo, Japan) and processed with FV10-ASW and ImageJ softwares.

**Immunohistochemical staining**. For the histopathological validation of the detection of phospho-S358 MLKL in muscles, immunohistochemical staining against a molecule of interest was performed in accordance with the original manner[72]. Formalin-fixed and paraffin-embed sections of muscle and renal samples were deparaffinized and heated in a pressure pot for 1 min in citrate buffer (10 mM, pH 6.0) to retrieve antigens. The slides were blocked with 10% goat serum and 0.1 M glycine in PBS, and then the sections were incubated with primary antibodies. The bound antibodies were visualized with peroxidase-labeled amino acid polymer-conjugated secondary antibodies and the associated substrate, diaminobenzene.

**Western blotting**. Cells were lysed with sodium dodecyl sulfate (SDS) lysis buffer (62.5 mM Tris-HCl, pH 6.8, 2.1% SDS, 15% glycerol) supplemented with protease inhibitors (cOmplete, mini, Sigma-Aldrich, St. Louis, Montana, USA). The samples were sonicated with Bioruptor UCD-250 (Cosmo Bio, Tokyo, Japan) followed by centrifugation at 4000 g for 5 min. Protein concentration in the supernatants was measured with Pierce 660 nm Protein Assay Kit (Thermo Fisher Scientific, Waltham, Massachusetts, USA). The samples were denatured with sample buffer (25 mM Tris-HCl (pH 6.5), 1% SDS, 5% glycerol, 0.05% Bromophenol Blue) in the presence of 5% (v/v) 2-mercaptoethanol and boiled for 5 min. 30 μg per lane of the total protein was fractionated with 5–20% SDS-PAGE gel (ATTO Corporation, Tokyo, Japan), and then transferred onto a polyvinylidene difluoride membrane using a transfer apparatus (Bio-rad, Hercules, California, USA). After blocking with 5% bovine serum albumin in TBST (10 mM Tris, pH 8.0, 150 mM NaCl, 0.5% Tween 20) for 30 min, the membrane was washed once with TBST and probed with primary antibodies overnight at 4 °C. The membrane was washed for 10 min three times with TBST, and incubated with secondary antibodies for an hour. The blots were washed three times with TBST, and developed with the ECL Prime Western Blotting Detection Reagent (GE Healthcare Life Sciences, Buckinghamshire, England). Protein bands were detected with LAS-4000 Imaging System (Fujifilm, Tokyo, Japan). Uncropped scans are shown in the Supplementary Fig. 8.

**Immunoblotting analysis of redox isoforms of HMGB1**. The serum and the muscle homogenate of CIM was immunoprecipitated using anti-HMGB1 antibodies and protein G magnetic beads (Tamagawa Seiki, Nagano, Japan) according to the manufacture's instructions. The immunoprecipitates and the supernatant of the co-culture were denatured with sample buffer (25 mM Tris-HCl (pH 6.5), 1% SDS, 5% glycerol, 0.05% Bromophenol Blue) in the presence or absence of 5% (v/v) 2-mercaptoethanol and boiled for 5 min. The samples were fractionated with 15% SDS-PAGE gel (ATTO Corporation, Tokyo, Japan) and transferred onto a polyvinylidene difluoride membrane. After the blocking and the incubation with

primary antibodies, the membrane was incubated with Clean-Blot IP detection Reagent (Thermo Fisher Scientific, Waltham, Massachusetts, USA) for an hour. The blots were developed with the ECL Prime Western Blotting Detection Reagent.

**Enzyme-linked immunosorbent assay (ELISA).** The levels of HMGB1 in the supernatant of the co-culture and the serum of CIM were measured with ELISA kit (Shino-test, Kanagawa, Japan) as described in the manufacturer's instructions. The levels of IL-1α, IL-1β, and IL-6 of the supernatant of the co-culture and the muscle homogenate of CIM were measured with Quantikine ELISA kit (R&D Systems, Minneapolis, Minnesota, USA) as described in the manufacturer's instructions.

**Statistics.** In time course analysis, the statistical significance was determined using Two-way analysis of variance (ANOVA) test followed by Dunnett's test or Log-rank test followed by Holm–Sidak multiple comparisons. Comparison between two groups was done using Mann–Whitney $U$ test for non-parametric analysis or with two-tailed Student's $t$-test for parametric analysis. Comparison among more than three groups was done using Kruskal–Wallis test followed by Dunn's test for non-parametric analysis or with one-way ANOVA test followed by Dunnett's test or Bonferroni post hoc test for parametric analysis. In the analysis of grip strength, two-way ANOVA test followed by Dunnett's multiple comparison test was used. All statistical tests were two-sided. PRISM (Ver.6, 7, and 8, GraphPad, CA, USA) was used for statistical analysis.

**Reporting summary.** Further information on research design is available in the Nature Research Reporting Summary linked to this article.

## Data availability
All the data supporting the findings from this study are available within the manuscript and its supplementary information. Source data are provided with this paper.

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

## Acknowledgements

We thank Dr. Naoko Okiyama for providing the splenocytes of mutant OT-I mice lacking PRF1 and GZMB, Katsuko Yamasaki for the histological analysis, Dr. Naoki Kimura, Dr. Natsuka Umezawa, and Dr. Hirokazu Sasaki for obtaining muscle specimens from the patients, Dr. Shinichi Uchida for encouraging the study, Mr. Huon Wong and Ms. Jacinta Hansen for technical assistance, Dr. James Murphy, Dr. John Silke, and Dr. Jo Hildebrand for providing *Ripk3*$^{-/-}$ and *Mlkl*$^{-/-}$ mice, and Dr. Hung Nguyen for helpful discussion. This work was supported by JSPS KAKENHI Grant Number JP19K23839 (to M.K.), JP19K08903 (to F.M.) and JP15H04863 (to H.K.), Bristol-Myers Squibb Foundation grant 41907503 (to F.M.), National Health and Medical Research Council of Australia 1113577 (to I.P.W.), and John T. Reid Charitable Trusts (to I.P.W.). I.P.W. is supported by a National Health and Medical Research Council Medical Research Future Fund Practitioner Fellowship (1154325). This study was made possible through Victorian State Government Operational Infrastructure Support and the Australian Government National Health and Medical Research Council Independent Research Institute Infrastructure Support scheme.

## Author contributions

M.K. and F.M. conceived of the project, designed the study. M.K., F.M., and S.Y. wrote the manuscript. M.K. and J.D. were responsible for the acquisition of data. M.K., F.M., K.K., D.W., M.N., J.D., C.L., I.P.W., H.K. and S.Y. performed the data analysis and the interpretation. F.M. supervised the work. H.K. and S.Y. co-supervised the work.

## Competing interests

F.M. received research funding from AbbVie, Astellas Pharma, Bristol-Myers Squibb, Chugai Pharmaceutical, Daiichi Sankyo Company, Eisai, Eli Lilly and Company, ImmunoForge, Japan Blood Products Organization, Mitsubishi Tanabe Pharma, Novartis Pharma Japan, Ono Pharmaceutical, Otsuka Pharmaceutical Factory, Pfizer, Sanofi, Takeda Pharmaceutical Company and Teijin, consulting fees from Asahi Kasei Pharma and ImmunoForge, and speaking fees from AbbVie, Asahi Kasei Pharma, Bristol-Myers Squibb, Chugai Pharmaceutical, Eizai, Eli Lilly and Company, Glaxo Smith Kline, Ono Pharmaceutical, and Pfizer. H.K. received consulting fees from CSL Behring and Japan Blood Products Organization. S.Y. received research funding from Abbvie, Asahi Kasei Pharma, Chugai Pharma, CSL Behring, Eisai, ImmunoForge, Mitsubishi Tanabe Pharma, and Ono pharmaceutical, speaking fees from Abbvie, Asahi Kasei Pharma, Chugai Pharmaceutical, Eisai, Eli Lilly, GlaxoSmithKline, Mitsubishi Tanabe Pharma, Ono pharmaceutical, and Pfizer. I.P.W. received consulting fees from CSL Limited. M.K., K.K., D.W., M.N., J.D., and C.L. declare that no conflict of interest exists.
