## [Peer Review File · Nature Communications]

Targeting necroptosis in muscle fibers ameliorates inflammatory myopathiesREVIEWER COMMENTS

Reviewer #1 (Remarks to the Author):

The authors present a series of experiments suggesting that the pattern of cell death of muscle fibers in PM is FAS 25 ligand-dependent necroptosis, while that of satellite cells and myoblasts is perforin 1/granzyme 26 B-dependent apoptosis, using human muscle biopsy specimens of PM patients and models of PM 27 in vitro and in vivo. They suggest the novel concept that targeting necroptosis in muscle cells is a promising strategy for treating PM providing an alternative to current therapies directed at leukocytes.

My comments and concerns are:

1. There is considerable controversy now as to the existence of the myositis syndrome known as polymyositis, as many of these subjects may in fact have different syndromes; thus, it would be useful to have more details as to the documentation of PM criteria and Tanimoto's criteria and particularly the clinical, serological and pathologic data supporting these diagnoses.
2. The tibialis anterior muscle is an unusual location for a myositis clinical biopsy and can the authors explain why this muscle was biopsied and if there is any information on whether this muscle differs from other muscles more commonly involved and biopsied in myositis patients.
3. While much of the focus in PM is on muscles, these are systemic diseases with frequent pulmonary, cardiac, joint and gastrointestinal involvement, which are not studied nor commented on to any extent in this paper. This point should be included, as well as any information about the roles of necroptosis, apoptosis and the possible implications of their work on other tissues besides muscle, especially relating to potential therapeutic potential or downsides of such therapy.
4. Given the relatively small number of samples being tested throughout the study, and some with inconsistent results, some of the statements regarding results should be more qualified.

Reviewer #2 (Remarks to the Author):

Thank you for letting me review the manuscript NCOMMS-20-29944 entitled "Targeting necroptosis in muscle fibers ameliorates inflammatory myopathies" by authors Dr Mizoguchi and co-workers.

In this paper the authors claim that the injured muscle fibers in the chronic inflammatory myopathy, polymyositis, release pro-inflammatory molecules, which would accelerate a CTL-induced muscle injury, and inhibition of the cell death of muscle fibers could be a novel therapeutic strategy to suppress both muscle injury and inflammation in PM. Using human muscle samples from patients, muscle cell lines and a mouse model for polymyositis, they could confirm their hypothesis and have demonstrated a possible new pathway to target for treatment in these patients.

The results are logically and clearly presented and the discussion is relevant. However, there are several issues that need to be addressed by the authors to make the paper and conclusions even more clear.

A weakness is the low number of muscle biopsy samples from patients.

I suggest that the authors add some more information on the patients: demographic data, and how the diagnosis of PM was defined more specifically, were these patients classified as probable or definite PM?

Did these patients have similar muscle histopathology?

I am a bit surprised to see the muscle histopathology in Figure 1B, to me the features would rather suggest a patient with anti-Jo1 autoantibodies or dermatomyositis. I suggest to add information on the phenotypes of infiltrating mononuclear cells in the muscle tissue of the included patients?

Did the included patients have any detectable autoantibodies?

Results:

The molecule HMGB1 exists in three different redox isoforms, with distinct functions,

proinflammatory and non inflammatory. For this study it would add important information to include the type of redox form of HMGB1 that was measured in the co-culture of myotubes as well as in the sera of the CIM model and also which form was expressed in the muscle fibers of the CIM model,

Fig 5 A. I suggest to mark the injured fibers in HE stainings as well as in the immunofluorescence staining

CFLAR staining in muscle fibers of the mice is difficult to see and does not look like the staining of the human fibers

Reviewer #3 (Remarks to the Author):

Kamiya et al. suggest that necroptosis, a recently identified pathway of regulated necrosis, contributes to muscle fiber degeneration in inflammatory myopathies. Several concerns preclude publication of the manuscript in the opinion of this referee.

1) To the best of the knowledge of this referee, no one has ever detected necroptosis (pMLKL staining) in human muscle fibers. This experiment should be performed in the patient MB samples in the original manner (immunohistochemistry, as published in PMID 28388412).

2) RIPK3-deficient and MLKL-deficient, or RIPK1-kinase dead mice must be investigated in the CIM model.

3) Other pathways of regulated necrosis (not granzyme or perforin which are involved in apoptosis) should be investigated in the same model. To start, and because it is easy to test, I recommend to include Fer-1 treatment (ferroptosis was suggested to contribute to muscle damage and heart attacks), and ideally, but not required, GSDMD-deficient mice (pyroptosis).

4) Necroptosis is thought to help humans defend viral infection. Treatment of inflammatory myopathies with Nec-1s over a long time would be likely to cause side effects. A discussion on how this could be clinically applicable is suggested.

POINT-BY-POINT RESPONSE

We wholeheartedly thank all reviewers for their helpful comments and constructive criticisms concerning our manuscript entitled “Targeting necroptosis in muscle fibers ameliorates inflammatory myopathies (No. NCOMMS-20-29944)”. Below, we have provided point-by-point responses to all of the reviewers’ comments and suggestions. The responses to the reviewers’ comments are written in **blue**, and the changed parts in the revised manuscript are written in **red**.

Response to Reviewer #1

Comment 1): There is considerable controversy now as to the existence of the myositis syndrome known as polymyositis, as many of these subjects may in fact have different syndromes; thus, it would be useful to have more details as to the documentation of PM criteria and Tanimoto’s criteria and particularly the clinical, serological and pathologic data supporting these diagnoses.

Response: We thank the reviewer for this comment. As the reviewer pointed out, inflammatory myopathy (IIM) consists of heterogeneous subsets with different clinical, serological and histopathologic features. We should have described the detailed information on patients listed above as well as how they met the classification criteria. We have added a list of these characteristics in Supplementary Table 1.

Comment 2): The tibialis anterior muscle is an unusual location for a myositis clinical biopsy and can the authors explain why this muscle was biopsied and if there is any information on whether this muscle differs from other muscles more commonly involved and biopsied in myositis patients.

Response: We agree with the reviewer that this should be explained. We have added the following sentences in the Material and methods section.

The conchotome muscle biopsy from the tibialis anterior muscle is a simple and less invasive procedure with a low complication rate, comparing to conventional muscle biopsy. Its validity for the histopathological evaluation of PM/DM has been well evaluated as the samples obtained from the tibialis anterior muscle of PM/DM patients with the procedure showed no significant difference in the frequency of inflammatory infiltrates, necrotic muscle fibers, and regenerating muscle fibers compared to those obtained from the proximal muscles of the same patients. All the patients included in the analysis showed necrotic muscle fibers, which were identified with reduced eosin staining in the cytoplasm but did not have the characteristic feature of immune-mediated

necrotizing myopathy. The patients who were suspicious of cancer-associated, viral, or immune check point inhibitor-associated myositis were excluded.

Comment 3) While much of the focus in PM is on muscles, these are systemic diseases with frequent pulmonary, cardiac, joint and gastrointestinal involvement, which are not studied nor commented on to any extent in this paper. This point should be included, as well as any information about the roles of necroptosis, apoptosis and the possible implications of their work on other tissues besides muscle, especially relating to potential therapeutic potential or downsides of such therapy.

Response: We thank the reviewer for raising this very important point. As the reviewer pointed out, IIMs are the systemic diseases with a variety of extramuscular involvements. Since there are no immunological animal models of IIM reproducing extramuscular organ involvement, we could not evaluate the effect of inhibition of necroptosis on the extramuscular organ involvement in vivo. Given the involvement of necroptosis in inflammatory diseases of various organs and efficacy of necroptosis inhibition against corresponding animal models, we expect necroptosis inhibition might also be an effective therapy against the extramuscular involvement of IIMs. We have added following paragraph in the Discussion section:

Inflammatory myopathies are the systemic diseases with a variety of extramuscular involvements, including the skin, lung, heart, joint, and gastro-intestinal tract. Since there are no immunological animal models of inflammatory myopathies reproducing these extramuscular involvements, the effect of inhibition of necroptosis on these organs could not be evaluated in vivo. Given the involvement of necroptosis in inflammatory diseases of various organs and the efficacy of necroptosis inhibitions in corresponding animal models such as dermatitis, acute respiratory distress syndrome, cardiomyositis, arthritis, and colitis, systemic inhibition of necroptosis could also be a potential therapy for the extramuscular involvement of inflammatory myopathies.

Comment 4) Given the relatively small number of samples being tested throughout the study, and some with inconsistent results, some of the statements regarding results should be more qualified.

Response: We thank the reviewer for this comment. We have increased the number of patients with IIM included in the analysis (9 PM patients and 3 DM patients) and added the list of their features in Supplementary Table 1. Regarding classification criteria, we have described Bohan and Peter's criteria and 2017 EULAR/ACR criteria together in order to categorize the patients with various characteristics.

Response to Reviewer #2

Comment 1): A weakness is the low number of muscle biopsy samples from patients. I suggest that the authors add some more information on the patients: demographic data, and how the diagnosis of PM was defined more specifically, were these patients classified as probable or definite PM?

Did these patients have similar muscle histopathology? I am a bit surprised to see the muscle histopathology in Figure 1B, to me the features would rather suggest a patient with anti-Jo1 autoantibodies or dermatomyositis. I suggest to add information on the phenotypes of infiltrating mononuclear cells in the muscle tissue of the included patients? Did the included patients have any detectable autoantibodies?

Response: We thank the reviewer for this comment. As the reviewer pointed out, inflammatory myopathy (IIM) consists of heterogeneous subsets with different clinical, serological and histopathologic features. We have increased the number of patients included in the analysis and added a list of clinical information in Supplementary Table 1. Regarding classification criteria, we have specified whether the patients were classified as definite or probable PM or dermatomyositis (DM) based on both Bohan and Peter's criteria and 2017 EULAR/ACR criteria. As the reviewer pointed out, the muscle histopathology originally shown in Figure 1B and 1C was of a patient who was positive for anti-Jo-1 antibodies (patient #3 in Supplementary Table 1). We have replaced Figure 1B and 1C with the histopathology (of patient #1 in Supplementary Table 1) with marked endomysial inflammatory cell infiltrates, which is a hallmark histopathology of PM. Additionally, we have described the information on the phenotypes of infiltrating mononuclear cells in the muscle specimen in Supplementary Table 1. Strikingly, we found that many molecules associated with necroptosis were expressed in necrotic muscle fibers uniformly in patients with inflammatory myopathies, even though they had a variety of clinical, serological, and histopathological features.

Comment 2): Results: The molecule HMGB1 exists in three different redox isoforms, with distinct functions, proinflammatory and non-inflammatory. For this study it would add important information to include the type of redox form of HMGB1 that was measured in the co-culture of myotubes as well as in the sera of the CIM model and also which form was expressed in the muscle fibers of the CIM model,

Response: We thank the reviewer for this comment. We analyzed the redox state of HMGB1 in the co-culture supernatant and in the serum and muscle homogenates of CIM mice by immunoblotting and utilizing the difference between the redox isoforms in electrophoretic mobility in nonreducing conditions. We found that HMGB1 in co-culture

supernatants was mainly in the disulfide form (Supplementary Figure 3) and that in the serum and muscle homogenates were in both the all-thiol and disulfide form (Supplementary Figure 8). We have added the following sentences in the Results section:

In the section of “Inhibition of necroptosis suppressed CTL-induced release of inflammatory mediators from myotubes”

While the levels of High Mobility Group Box 1 (HMGB1), which is one of the family of DAMPs, and inflammatory cytokines including IL-1 α and IL-6, were not detectable in the supernatants of mono-cultured H2K^bOVA-myotubes, the levels of HMGB1, IL-1 α and IL-6 were increased in the co-culture of myotubes and OT-I CTLs (Figure 4G-I).

HMGB1 has different redox isoforms with distinct functions. The all-thiol HMGB1 and disulfide HMGB1 exert chemoattractant and cytokine activity, respectively. The redox state of HMGB1 in the supernatants was analyzed by immunoblotting and utilizing the difference in electrophoretic mobility under nonreducing conditions. While the all-thiol HMGB1 was detected as a single band with an apparent molecular weight of 28 kD both in reducing and nonreducing conditions, the disulfide HMGB1 was detected as a single band in reducing conditions with an apparent molecular weight of 26 kD but shifted in reducing conditions to 28 kD. According to the electrophoretic pattern, HMGB1 in the supernatants of the co-culture of myotubes and OT-I CTLs was in the disulfide form (Supplementary Figure 3). Inhibition of necroptosis in the myotubes by pretreating them with the inhibitor Nec1s prior to the co-culture suppressed the levels of these inflammatory mediators in the co-culture (Figure 4G-I).

In the section of “HMGB1 mediates muscle inflammation”

We hypothesized that the therapeutic effect of necroptosis inhibition on muscle inflammation was mediated by reducing the release of DAMPs from dying muscle fibers. We found that the level of HMGB1 in the serum was markedly increased in CIM (Figure 6A). Strikingly, treatment with the necroptosis inhibitor Nec1s suppressed CIM-induced increase of HMGB1 by about 90%. Consistent with a previous study of human PM²⁵, HMGB1 was highly expressed in the muscle fibers in the areas where inflammatory infiltrates were observed, indicating that the muscle fibers are a dominant source of HMGB1 in CIM (Figure 6B). Next, we investigated the redox state of HMGB1 by immunoprecipitating HMGB1 from the muscle homogenates and the serum of CIM. Majority of HMGB1 in both the muscle homogenates and the serum of CIM was in the all-thiol form and the levels of both the all-thiol and disulfide form was increased in the samples of the CIM mice compared to mice without CIM. Of note, the disulfide form of

HMGB1 was barely observed in the samples of the mice without CIM (Supplementary Figure 8A, B). To confirm if the increased release of HMGB1 is involved in the muscle inflammation, CIM mice were treated with anti-HMGB1 antibodies.

The partially inconsistent results between in vitro and in vivo might be due to the differences between the models or the contribution of HMGB1 derived from other cell types such as monocytes or macrophages in vivo. In addition, once HMGB1 is released from a cell, it remains unclear how HMGB1 is oxidized or reduced and how it acts in vivo over time and space.

Although it might help to further characterize the redox isoforms of HMGB1 by conducting a quantitative mass-spectrometry analysis, this method has produced some controversial results which have led to multiple retractions (Yang H, et al. Mol Med 2012, Palmblad K, et al. Mol Med 2015, Antoine DJ, et al. J Hepatol 2012, and so on). In light of this, we have shied away from conducting a mass-spectrometry analysis.

Comment 3): Fig 5 A. I suggest to mark the injured fibers in HE stainings as well as in the immunofluorescence staining

Response: We thank the reviewer for pointing this out. We have replaced the pictures shown in Figure 5A with those showing injured muscle fibers more clearly and marked the injured fibers in both HE staining and immunofluorescence staining.

Comment 4): CFLAR staining in muscle fibers of the mice is difficult to see and does not look like the staining of the human fibers

Response: We thank the reviewer for pointing this out. We have replaced the pictures shown in Figure 5A with those showing injured muscle fibers more clearly. Regarding the immunofluorescence staining against CFLAR, the replacement picture shows that the expression of CFLAR is upregulated in the injured muscle fibers, which was consistent with the finding of human muscle specimens.

Response to Reviewer #3

Comment 1): To the best of the knowledge of this referee, no one has ever detected necroptosis (pMLKL staining) in human muscle fibers. This experiment should be performed in the patient MB samples in the original manner (immunohistochemistry, as published in PMID 28388412.

Response: We thank the reviewer for this comment. We conducted the immunohistochemical staining against pMLKL on muscle specimens of the patients according to the original manner (Gong YN, et al. Cell 2017) and human kidney

samples were used as a negative control and have included the results in Supplementary Figure 1C. Because of the scarcity of the samples which were obtained by conchotome muscle biopsy and that muscle specimens are conventionally analyzed as frozen sections, we could only conduct the immunohistochemical staining of formalin-fixed and paraffin-embed samples of the muscles from a small number of patients (n = 3) and the representative images were shown in Fig S1.

Comment 2): RIPK3-deficient and MLKL-deficient, or RIPK1-kinase dead mice must be investigated in the CIM model.

Response: We thank the reviewer for this comment. We analysed the effect of RIPK3 or MLKL deficient mice on CIM and found that the histological scores and necrotic muscle areas were ameliorated in the muscles of *Ripk3*^{-/-} or *Mkl1*^{-/-} mice compared to those of wild type mice (Supplementary Figure 5A-C). We have added the following sentences in the Results section:

In the section of “Inhibition of necroptosis ameliorated muscle strength and inflammation in a murine model of PM”

Next, we analyzed the effect of the deficiency of RIPK3 or MLKL in CIM. In both *Ripk3*^{-/-} and *Mkl1*^{-/-} CIM mice, the histological inflammation scores (Supplementary Figure 5A, C) as well as the necrotic areas (Supplementary Figure 5B, D) in the muscles decreased compared to those of the wild-type CIM mice, implying the involvement of necroptosis in the pathophysiology of CIM.

Comment 3): Other pathways of regulated necrosis (not granzyme or perforin which are involved in apoptosis) should be investigated in the same model. To start, and because it is easy to test, I recommend to include Fer-1 treatment (ferroptosis was suggested to contribute to muscle damage and heart attacks), and ideally, but not required, GSDMD-deficient mice (pyroptosis).

Response: We thank the reviewer for this suggestion. We examined the therapeutic effect of the inhibition of ferroptosis and pyroptosis using small molecule inhibitors, ferrostatin, VX765, and disulfiram. We found none of these inhibitors ameliorated CIM-induced muscle weakness nor histological inflammation (Supplementary Figure 7). We have added following sentences in the section of *“Inhibition of necroptosis ameliorated muscle strength and inflammation in a murine model of PM”* in the Result:

Additionally, we examined the therapeutic effect of the inhibition of ferroptosis and pyroptosis, other types of regulated form of cell death with the necrotic morphological features. Treatment with the inhibitor of ferroptosis, ferrostatin-1 (Fer-1) or the inhibitors of pyroptosis, belnacasan (Vx765) or disulfiram did not ameliorate CIM-induced decrease of grip strength (Supplementary Figure 7A), the histological inflammation scores

(Supplementary Figure 7B), or the necrotic areas (Supplementary Figure 7C), suggesting these cell death pathways were less involved in the pathogenesis of CIM.

Comment 4): Necroptosis is thought to help humans defend viral infection. Treatment of inflammatory myopathies with Nec1s over a long time would be likely to cause side effects. A discussion on how this could be clinically implicable is suggested.

Response: We thank the reviewer for raising this very important point. We now realize that we should have clearly stated the potential adverse effects of the therapy targeting necroptosis. We have added following sentences in the Discussion setion:

While the physiological relevance of necroptosis remains largely unclear, necroptosis is indicated to be involved the inflammatory response in infectious diseases and tumor immunity. For the clinical application, it is necessary to take these potential side effects of long-term inhibition of necroptosis into consideration.

Once again, we thank all the reviewers for their helpful comments. We hope that the many new experiments carried out and described here both significantly improve the study and overall address the reviewer comments.

REVIEWER COMMENTS

Reviewer #1 (Remarks to the Author):

The revisions addressed my major concerns.

Reviewer #2 (Remarks to the Author):

It is with great interest I read the revised version of this manuscript which has improved. I find that the authors have responded to my comments and revised the manuscript in a satisfactory way. There are only a few minor comments and suggestions:

Figure 1C: difficult to see the green and the red staining by immunofluorescence, I suggest to include new figures.

Supplementary Figure 1D staining for Caspase 8 18 kDa is not possible to see in some patients where there are arrows eg pat 3 and 5.

Reviewer #3 (Remarks to the Author):

1) The authors have added information of RIPK3- and MLKL-deficient mice to the study. They also looked at expression of the p345 (abcam) MLKL antibody, but unfortunately did not control this antibody with now available MLKL-ko mice. This particular antibody is known to cause off-target bands in Western blots that might well cause non-specific stainings in IF.

2) The "dying muscle fibre" in Fig. 5 is not at all swelling. How is this in line with necroptosis?

3) The IF provided does not indicate that muscle fibres are positive for pMLKL, but unlike surrounding tissue, the fibres rather do not stain positive at all. The same is true for the IHC presented in Figure S1. It looks as though lots of tissues surrounding the fibres are positive, but the fibres themselves are not!

4) Figure S5 is the strongest single piece of evidence in support of the title. It should be transferred to a figure within the manuscript rather than the supplement. The necrotic fibres were quantified. They should be demonstrated as well.

5) Figure S7 adds an important piece to this manuscript and convincingly rules out a major involvement of ferroptosis.

POINT-BY-POINT RESPONSE

We wholeheartedly thank all reviewers for their helpful comments and constructive criticisms concerning our manuscript entitled “Targeting necroptosis in muscle fibers ameliorates inflammatory myopathies (No. NCOMMS-20-29944A)”. Below, we have provided point-by-point responses to all of the reviewers’ comments and suggestions. The responses to the reviewers’ comments are written in blue, and the changed parts in the revised manuscript are written in red.

Response to Reviewer #1

Comment: The revisions addressed my major concerns.

Response: We thank the reviewer for this comment.

Response to Reviewer #2

Comment 1): It is with great interest I read the revised version of this manuscript which has improved. I find that the authors have responded to my comments and revised the manuscript in a satisfactory way. There are only a few minor comments and suggestions:

Figure 1C: difficult to see the green and the red staining by immunofluorescence, I suggest to include new figures.

Response: We thank the reviewer for pointing this out. We have replaced the pictures shown in Figure 1C with those showing red and green fluorescence more properly.

Comment 2): Supplementary Figure 1D staining for Caspase 8 18 kDa is not possible to see in some patients where there are arrows eg pat 3 and 5.

Response: We thank the reviewer for the suggestion. We have replaced the pictures of the patients such as #3, 5, 6, 7, 9, and 11 shown in Supplementary Figure 1D with those showing the immunofluorescence staining against active 18 kDa Caspase 8 subunit more clearly.

Response to Reviewer #3

Comment 1): The authors have added information of RIPK3- and MLKL-deficient mice to the study. They also looked at expression of the p345 (abcam) MLKL antibody, but unfortunately did not control this antibody with now available MLKL-ko mice. This particular antibody is known to cause off-target bands in Western blots that might well cause non-specific stainings in IF.

Response: We agree with the reviewer that this should be shown. We re-conducted the immunofluorescence staining against MLKL and phosphorylated-S345 MLKL (phospho-S345 MLKL) on muscle specimens of CIM using those of MLKL-deficient CIM mice as negative control. As shown in the replaced Figure 5A and newly added Supplementary Figure 4C, we could confirm that immunofluorescence on the dying muscle fibers of CIM was not due to non-specific binding of these antibodies.

Comment 2): The "dying muscle fibre" in Fig. 5 is not at all swelling. How is this in line with necroptosis?

Response: We thank the reviewer for this comment. While the inflamed muscle fibers often undergo atrophy resulting in a reduction in their size (Dorph C, et al. *Ann Rheum Dis* 2006, Cai C, et al. *Mod Pathol* 2019), some are enlarged. We speculate that there might be variations in the size of the dying muscle fibers depending on the affects and the time course of injury and cell death. It is true that necroptosis is morphologically characterized by cell swelling, we have replaced Figure 5A, B, and Figure 6B with those of the dying muscle fibers with cell swelling to avoid misleading. We have also shown the pictures of the dying muscle fiber with the feature of atrophy as Supplementary Figure 4A and B and added the following sentences in the results:

In the section of *"Inhibition of necroptosis ameliorated muscle strength and inflammation in a murine model of PM"*

Also, expression of RIPK1, RIPK3, MLKL, phosphorylated-S345 MLKL, FAS, CFLAR, and FAS was detected in the injured muscle fibers in CIM (Figure 5B, **Supplementary Figure 4B, C**), while these molecules were not observed in non-injured muscle fibers (Supplementary Figure 4D). **While necroptosis is morphologically characterized by cell swelling, some of the injured muscle fibers of CIM were rather shrunk, possibly representing atrophy induced by inflammation^{21,22} (Supplementary Figure 4A, B).** These observations were consistent with the histological findings of human PM.

Comment 3): The IF provided does not indicate that muscle fibres are positive for pMLKL, but unlike surrounding tissue, the fibres rather do not stain positive at all. The same is true for the IHC presented in Figure S1. It looks as though lots of tissues surrounding the fibres are positive, but the fibres themselves are not!

Response: We thank the reviewer for this comment. We re-evaluated the samples and found that there was variation in the localization of phosphorylated-S358 MLKL (pMLKL) among the dying muscle fibers, with some showing uniform expression throughout the cytoplasm and others showing marked expression on the plasma membrane. The translocation of pMLKL on plasma membrane increases upon the execution of necroptosis (Rodriguez D, et al. *Cell Death Differ* 2016). Hepatocytes,

which are multinucleated large cells as well as muscle fibers, show marked localization of pMLKL on their plasma membrane in hepatitis (Günther G, et al. *J Clin Invest* 2016, Samson AL, et al. *Nat Commun* 2020). Accordingly, we speculate that the variations in pMLKL localization in the cell might depend on the time course of necroptosis and cell type. We have replaced the pictures in Figure 1B and Supplementary Figure 1B and added some pictures of the other patients in Supplementary Figure 1C to more properly show the expression of pMLKL and variations in its localization. Accordingly, we have added the following sentences in the results:

In the section of “Cell death of muscle fibers is necroptotic in PM”

Immunofluorescence staining revealed that the dying muscle fibers, which are identified as cells with reduced eosin staining in the cytoplasm, expressed RIPK1, RIPK3, MLKL and phosphorylated MLKL, whereas the expression levels of these proteins in the intact muscle fibers were low or absent (Figure 1B, Supplementary Figure 1B, C). **We found some of the dying muscle fibers expressed phosphorylated MLKL, which localizes on the plasma membrane upon necroptosis¹³⁻¹⁵. Some of such muscle fibers expressed pMLKL throughout cytoplasm while others did markedly on their plasma membrane, suggesting there could be variations in its localization among the muscle fibers depending on the execution status of necroptosis.** While expression of CASP8 was observed in both PAX7 positive satellite cells and dying muscle fibers, expression of the active 18 kDa CASP8 subunit was only observed in the satellite cells (Figure 1C, Supplementary Figure 1D).

Comment 4): Figure S5 is the strongest single piece of evidence in support of the title. It should be transferred to a figure within the manuscript rather than the supplement. The necrotic fibres were quantified. They should be demonstrated as well.

Response: We thank the reviewer for this suggestion. We have relocated Supplementary Figure 5A-D to Figure 5C-F in the main manuscript.

Comment 5): Figure S7 adds an important piece to this manuscript and convincingly rules out a major involvement of ferroptosis.

Response: We thank the reviewer for this comment.

Once again, we thank all the reviewers for their helpful comments and suggestions. We hope that the additional experiments and evaluations carried out and described here both significantly improve the study and overall address the reviewer comments.

REVIEWERS' COMMENTS

Reviewer #3 (Remarks to the Author):

The authors have adequately adressed all concernes raised by this referee. I have no further comments and recommend publication.